# Effects of Dietary Inclusion of Sericea Lespedeza Hay on Feed Intake, Digestion, Nutrient Utilization, Growth Performance, and Ruminal Fermentation and Methane Emission of Alpine Doelings and Katahdin Ewe Lambs

**DOI:** 10.3390/ani12162064

**Published:** 2022-08-13

**Authors:** Wei Wang, Amlan Kumar Patra, Ryszard Puchala, Luana Ribeiro, Terry Allen Gipson, Arthur Louis Goetsch

**Affiliations:** 1College of Animal Science and Veterinary Medicine, Shenyang Agricultural University, Shenyang 110866, China; 2American Institute for Goat Research, Langston University, Langston, OK 73050, USA; 3Department of Animal Nutrition, West Bengal University of Animal and Fishery Sciences, Kolkata 700037, India

**Keywords:** condensed tannins, methane, small ruminants

## Abstract

**Simple Summary:**

Ruminants contribute to global greenhouse gas production, mainly through enteric methane emission (EME). Different dietary strategies have been explored to mitigate EME, including the use of forages rich in tannins that inhibit methanogenic activity in the rumen. Feeding the condensed tannin-containing forage Sericea lespedeza (*Lespedeza cuneata*) has reduced EME by small ruminants in previous studies. In the present experiment, the digestion of fiber, and in particular of nitrogen, decreased as the concentration of lespedeza increased. Similar metabolizable energy intake and retained energy among diets suggests that decreasing digested nitrogen intake and protein status were responsible for marked declines in growth performance as the dietary amount of lespedeza increased. In contrast to many other reports, the dietary concentration of lespedeza did not affect EME by goats or sheep under these feeding conditions.

**Abstract:**

Twenty-four Alpine doelings, initial 25.3 ± 0.55 kg body weight (BW) and 10.4 ± 0.11 mo of age, and 24 Katahdin ewe lambs, 28.3 ± 1.02 kg BW and 9.6 ± 0.04 mo of age, were used to determine effects of dietary inclusion of Sericea lespedeza (*Lespedeza cuneata*) hay on feed intake, digestion, growth performance, energy metabolism, and ruminal fermentation and methane emission. There were four periods, the first three 42 days in length and the fourth 47 days. Diets consumed ad libitum contained 75% coarsely ground hay with alfalfa (ALF), a 1:1 mixture of ALF and LES (ALF+LES), and LES (10.0% condensed tannins; CT). The intake of dry matter (DM) tended to be greater (*p* = 0.063) for Katahdin than for Alpine (4.14 vs. 3.84% BW; SEM = 0.110). The dry matter intake was similar among the diets (3.97, 4.10, and 3.89% BW for ALF, ALF+LES, and LES, respectively; SEM = 0.134). The digestion of organic matter (75.3, 69.3, and 65.5%; SEM = 0.86), neutral detergent fiber (61.7, 50.5, and 41.4%; SEM = 1.49), and nitrogen (78.8, 66.9, and 50.8% for ALF, ALF+LES, and LES, respectively; SEM = 0.92) decreased as the dietary concentration of lespedeza increased (*p* < 0.05). However, there was an interaction (*p* < 0.05) between the breed and diet in nitrogen digestion, with a greater value for goats vs. sheep with LES (54.4 vs. 47.3%; SEM = 1.30). The digested nitrogen intake decreased markedly with the increasing quantity of lespedeza (38.0, 27.5, and 15.7 g/day for ALF, ALF+LES, and LES, respectively; SEM = 1.26). The average daily gain was greater for Katahdin than for Alpine (*p* < 0.001; 180 vs. 88 g, SEM = 5.0) and ranked (*p* < 0.05) ALF > ALF+LES > LES (159, 132, and 111 g, respectively; SEM = 6.1). The ruminal methane emission differed (*p* < 0.05) between animal types in MJ/day (1.17 and 1.44), kJ/g DM intake (1.39 and 1.23), and kJ/g ADG (18.1 and 9.8 for Alpine and Katahdin, respectively). Regardless of the period and animal type, diet did not impact methane emission in MJ/day or relative to DM intake, BW, or ADG (*p* > 0.05). The digestible and metabolizable energy intakes, heat production, and retained energy were not affected by diet (*p* > 0.05). In conclusion, future research should consider the marked potential effect of CT of forages such as lespedeza on nitrogen digestion and associated effects on protein status and other conditions that may be impacted.

## 1. Introduction

Sericea lespedeza (*Lespedeza cuneata*) is a perennial legume adapted to subtropical and temperate climates, which may be used for grazing, hay, and silage for ruminants [1,2]. Lespedeza is lower in nutritive value than alfalfa because of higher concentrations of total fiber, lignin, and bioactive condensed tannins (CT) [1,3]. However, studies over the last 15–20 years have demonstrated the effectiveness of this forage to reduce gastrointestinal parasitic infections and prevent bloat in ruminants due to the presence of CT [1,4]. This has contributed to a renewed interest in the use of lespedeza as a medium-quality warm-season nutraceutical forage for livestock in different parts of the world [1,5]. 

Ruminants make a significant contribution to the emission of the greenhouse gas methane [6,7]. Therefore, several methane mitigation strategies, including chemical, dietary, breeding practices, and vaccination, have been studied [7,8]. Among them, the use of dietary phytochemicals such as CT, saponins, and essential oils, being natural components of many feedstuffs and forages, is of special interest [9]. The presence of CT in lespedeza has been exploited in a number of studies to reduce methane production in goats and sheep [10,11,12,13,14,15].

Goats and sheep differ in many ways when fed tannin-rich diets and forage, including the level of feed intake, feeding behavior, nutrient digestibility, and growth performance. Compared with sheep, due to their evolutionary lineage as browsing animals, goats are considered more tolerant and adaptive to forage and browse plant species containing high concentrations of CT [16,17]. It has been reported that CT in lespedeza did not affect dry matter (DM) intake in goats but reduced DM intake and digestibility in sheep [18]. The adaptation of goats to the consumption of tannins has been attributed to the secretion of tannin-binding proteins in saliva, higher rates of tannin degradation by ruminal microbes, and greater recycling of urea [16,18]. Goats and sheep could differ in other responses as well, such as growth performance, nutrient digestibility, fatty acid profile in milk, and ruminal fermentation conditions and microbiota composition [19]. Therefore, this study was conducted to investigate the effects of dietary concentrations of lespedeza and alfalfa on feed intake, digestion, nutrient utilization, growth performance, and ruminal fermentation conditions and methane emission by Alpine doelings and Katahdin ewe lambs. 

## 2. Materials and Methods

### 2.1. Animals, Periods, and Housing

The protocol for the experiment was approved by the Langston University Animal Care and Use Committee. Twenty-four Alpine doelings (ALP; initial body weight and age of 25.3 ± 0.55 kg and 10.4 ± 0.11 mo, respectively) and twenty-four Katahdin ewe lambs (KAT; 28.3 ± 1.02 kg and 9.6 ± 0.04 mo, respectively) were used. The treatment arrangement was a 2 × 3 factorial, with the two species or breeds and three diets, and the experiment entailed a completely randomized design. The ALP and KAT were allocated to the three dietary treatments (*n* = 8 for each breed) for similar mean and variation in body weight (BW) and age. At most times, animals resided in six (one for each breed per diet) 6.1 m × 5.6 m pens in an enclosed building that had a 6.1 m × 1.35 m area with a concrete floor and a 6.1 m × 4.25 m unpaved floor area. The pens included Calan gate feeders (American Calan, Inc., Northwood, NH, USA) for individual feeding. There was a 2-wk preliminary period for training in use of Calan gate feeders. The study began in January 2019 and the duration was 173 days, with the first three periods 6 wk in length and the fourth 47 days. The animals consumed the treatment diets continuously for the total period of 173 days. Ambient temperature and relative humidity were determined every 30 min with three Hobo^®^ Temperature/RH Data Loggers (model number U12-011; Onset Computer Corp., Bourne, MA, USA) placed in different areas of the facility.

The pens and feeders were aligned in a row adjacent to one another. With diets potentially differing in palatability, differences between breeds in size and behavior, and similar environmental conditions among pens, the same diet was fed to animals of the same breed in pens to avoid problems with attempts to gain access to feeders other than their own. Thus, each pen hosted a different animal type consuming a different diet. Animals within pens and treatment groups were randomly assigned to two sets, with set 2 beginning the experiment 1 wk after set 1. During the last 7 days of each period, animals were housed in 0.7 m × 1.2 m metabolism crates. The first 2 days were for adaptation, with feces and urine collections on the subsequent 5 days. On four of these latter days, animals were cycled in groups of six into a room with metabolism crates fitted with headboxes of a respiration calorimetry system for 1 day of measures.

### 2.2. Diets

Diets were complete mixtures of 25% concentrate and 75% coarsely ground forage (Table 1) and were fed at 08:00 h after collecting and weighing refusals. Dietary forage was all alfalfa hay (ALF), a 1:1 mixture of alfalfa and Sericea lespedeza hay (ALF+LES), and all lespedeza hay (LES). The variety of LES was AU Grazer. The diets were offered at approximately 110% of consumption on the preceding few days. In addition, there was a trace mineral salt block in each pen (Big 6 Mineral Salt, American Stockman, Overland Park, KS, USA; 96.5–99.5% NaCl, 4000 mg/kg Zn, 1600 mg/kg Fe, 1200 mg/kg Mn, 260–390 mg/kg Cu, 100 mg/kg I, and 40 mg/kg Co; as-fed basis), and small pieces were available in the bottom of feeders when animals were in metabolism cages.

### 2.3. Measures

#### 2.3.1. Feed Intake and Growth Performance

BW was determined at the beginning of the experiment, the end of each period, and at the start and end of calorimetry measurement days. Average daily gain (ADG) in each period was determined from initial and final BW. The Kleiber ratio [20] for the study was estimated as the ratio of average BW gain to the mid-point kg BW^0.75^ [21].

#### 2.3.2. Digestibility, Metabolizability, Energy Utilization, and Methane Emission

Feed was sampled once weekly. Urine was acidified with 30 mL of 30% (*v*/*v*) H_2_SO_4_ placed in collection vessels to maintain pH below 3.0, and composite samples of feces and urine were formed by collecting 15% daily aliquots. Partial DM concentration in feces was determined by drying in a forced-air oven at 55 °C for 48 h. Fecal samples were analyzed for DM (100 °C), ash [22], nitrogen (N; LecoTruMac CN, St. Joseph, MI, USA), neutral detergent fiber (NDF) using heat stable amylase [23], and containing residual ash, and gross energy (GE) using a bomb calorimeter (Parr 6300; Parr Instrument Co., Inc., Moline, IL, USA). Feed samples were analyzed at Custom Laboratory (Monett, MO, USA; customaglabs.com) for the same constituents by similar procedures, except for N that was determined by the Kjeldahl procedure [22]. In addition, concentrations of acid detergent fiber (ADF) and acid detergent lignin (ADL) were determined [23]. Urine samples were analyzed for DM (lyophilization), N (LecoTruMac CN), and GE using the procedures stated above. Digestible energy (DE) intake was estimated assuming 19.33 kJ/g digestible organic matter (OM) intake [24], with GE intake based on DE intake and OM digestibility. Two composite feed samples for each period were constructed from weekly samples for analysis of CT [25], using CT extracted from lespedeza as the standard.

Emission of methane and carbon dioxide and oxygen consumption were measured with an indirect, open-circuit respiration calorimetry system (Sable Systems International, North Las Vegas, NV, USA) with six metabolism crates and head boxes as described earlier [26,27]. Oxygen concentration was analyzed using a fuel cell FC-1B oxygen analyzer (Sable Systems International) and methane and carbon dioxide concentrations were measured with infrared analyzers (CA-1B for carbon dioxide and MA-1 for methane; Sable Systems International). Prior to gas exchange measurements, analyzers were calibrated with gases of known concentrations. Ethanol combustion tests were performed to ensure complete recovery of oxygen and carbon dioxide produced with the same flow rates as used during measurements. Heat energy (HE) was calculated from oxygen consumption and production of carbon dioxide and methane according to the Brouwer [28] equation without consideration of urinary N.

Digestibilities were based on feed intake 2 days before and the first 4 days of feces and urine collection. For energy measures, GE digestibility was applied to GE intake on the 2 days before and the day of calorimetry measures. Likewise, urinary energy as a percentage of GE intake during the period of feces and urine collection was applied to GE intake of calorimetry measures to determine relevant urinary energy output. Energy loss from ruminal methane emission was based on an energy concentration of 39.5388 kJ/L [28]. Intake of metabolizable energy (ME) was estimated as the difference between DE intake and the sum of energy in urine and methane. Recovered or retained (RE) was the difference between ME intake and HE.

#### 2.3.3. Ruminal Fluid and Blood Characteristics

Ruminal fluid was sampled by stomach tube on the middle day of wk 5 of each period at 4 h after feeding. The pH was measured with a digital meter and then 4-mL samples were placed into a tube with 1 mL of a 250 g/L metaphosphoric acid solution and frozen at −20 °C for later volatile fatty acid (VFA) analysis. Likewise, 3-mL samples were placed into a tube with 2 mL of 3 *M* HCl and frozen at −20 °C for ammonia N (AMN) analysis. Analyses of VFA and AMN were conducted by procedures of Lu et al. [29] and Broderick and Kang [30], respectively. For protozoa enumeration [31], 1 mL of ruminal fluid was combined in a tube with 4 mL of a methyl green, formalin, and saline solution (0.06 g methyl green, 0.85 g sodium chloride, 10 mL of 70% (*v*/*v*) formaldehyde solution, and 90 mL deionized water), followed by use of a 0.1 mm deep Neubauer hemocytometer counting chamber (Hausser Scientific, Horsham, PA, USA). Blood samples were collected at this time as well, centrifuged at 3000× *g* for 20 min at 4 °C to harvest plasma, stored frozen at −20 °C, and later thawed and analyzed for urea N (UN) concentration [32] and total antioxidant capacity (TAC) colorimetrically with a Technicon Autoanalyzer II System (Technicon Instruments, Tarrytown, NY, USA) based on a ferric reducing ability of plasma [33].

### 2.4. Statistical Analyses

Data were analyzed using mixed effects models with SAS [34,35]. For variables with values in different periods, fixed effects were breed, dietary treatment, period, and all interactions, with the repeated measure of period and random effect of animal within breed and dietary treatment. Because in many cases there is interest in overall differences and treatment effects during the entire time of implementation, main effect means of breed and(or) dietary treatment are presented in some instances even though an interaction involving period was significant. When the three-way interaction was significant, the analysis was conducted by period. Period was removed from the model for variables with one value during the experiment. Mean separation was through least significant difference with a protected F-test.

## 3. Results

### 3.1. Diet Composition and Environmental Conditions

The dietary CP concentration decreased considerably as the concentration of alfalfa decreased and that of lespedeza increased (Table 1). The intent was to compare different concentrations only of these two forage sources rather than to vary those of other ingredients in order to have similar dietary concentrations of constituents such as CP, ruminally degraded and undegraded protein, fiber fractions, etc. The concentrations of fiber fractions in ALF+LES were slightly greater than expected based on those in ALF and LES, which could relate to unrepresentative sampling or perhaps influence of CT on the analyses as has been previously noted [36]. Based on the CT concentration in LES, it would appear that the concentration in lespedeza hay was near 13%, which is fairly similar to concentrations of 14–20% noted in previous studies at this location [3,10,12,37] and greater than 5.5 and 8.4% in lespedeza recently used by Puchala et al. [38] and Liu et al. [39], respectively. Temperature, humidity, and temperature–humidity index (THI) values in Table 2 indicate that animals were not subjected to appreciable cold or heat stress, although it is notable that the THI increased by 8.2 and 9.6 units from period 2 to 3 and 3 to 4, respectively. Such factors may have contributed to the many interactions involving period.

### 3.2. Feed Intake and Growth Performance

Three-way interactions between breed, diet, and period were not significant for any variable (*p* > 0.05) other than ruminal fluid concentrations of some constituents as addressed later. BW was affected by breed, period, breed × period, and diet × period (*p* < 0.05; Table 3). BW increased as period advanced but the change for KAT was greater than for ALP (Table 4). For the diet × period interaction, BW was similar among diets in periods 1 and 2. Conversely, BW was greater for ALF than for LES in periods 3 and 4 (*p* < 0.05), with a greater difference in the final period (i.e., 6.6 vs. 4.8 kg).

DM intake in g/day was influenced by breed, period, and breed × period (*p* < 0.05; Table 3). For both ALP and KAT, values were higher in periods 2, 3, and 4 than in period 1, and the difference between breeds was less in period 1 than in other periods (Table 4). DM intake expressed in % BW was affected only by period, being greatest in period 2 and lowest in period 4 (*p* < 0.05). DM intake in g/kg BW^0.75^ differed among periods similarly, although there was a breed difference (*p* < 0.05).

ADG was affected by breed, diet, period, and diet × period (*p* < 0.05; Table 3), being greater for KAT than for ALP (Table 4). The overall diet main effect mean ranking was ALF > ALF+LES > LES (*p* < 0.05). ADG:DMI was affected (*p* < 0.05) by diet (ALF > ALF+LES and LES), breed (KAT > ALP), and a breed × period interaction. The ratio was greater (*p*< 0.05) for KAT vs. ALP in periods 1, 2, and 4 but not in period 3. The Kleiber ratio was greater for KAT than for ALP and ranked ALF > ALF+LES > LES (*p* < 0.05).

### 3.3. Digestibility and Nitrogen Balance

During the time of feces and urine collection, DM intake (g/day) was greater for KAT than for ALP, but as noted for the entire experiment, there was an interaction between breed and period (*p* < 0.05; Table 5 and Table 6). DM intake was similar among periods for ALP, but for KAT it was lower in period 1 than in periods 2 and 4 and greatest in period 3 (*p* < 0.05). The digestibilities of DM, OM, and NDF were affected (*p* < 0.05) by diet (ALF > ALF+LES > LES) and period (1 > 2, 3, and 4), but the values were similar between breeds. The intakes of digested DM, OM, and NDF were higher for KAT vs. ALP, and the digested NDF intake was greater for ALF than for ALF+LES and LES (*p* < 0.05).

The nitrogen intake was greater for KAT vs. ALP (*p* < 0.05), but there was a breed × period interaction, with similar values among periods for ALP but for KAT a ranking of period 1 < 2 and 4 < 3 (*p* < 0.05; Table 5 and Table 6). The digestibility of N was slightly greater for ALP than for KAT (2.3 percentage units; *p* < 0.05). An interaction between breed and diet (*p* < 0.05) was due to similar values between breeds with ALF and ALF+LES vs. a much greater value for ALP than for KAT with LES (7.1 percentage units; *p* < 0.05). This corresponded to a similar digested N intake between breeds with LES but greater values for KAT with other diets (*p* < 0.05).

Urinary N excretion, N retention, and urinary energy loss were higher for KAT than for ALP, and the diet ranking was ALF > ALF+LES > LES (*p* < 0.05; Table 5 and Table 6). ALP excreted less urinary N in period 1 than in periods 2, 3, and 4, whereas the excretion for KAT was the lowest in period 1 and highest in periods 3 and 4 (*p* < 0.05). DE intake also was higher for KAT than for ALP (*p* < 0.05).

### 3.4. Energy Measures during the Calorimetry Period

The results for measures such as the intake of DM and DE and urinary energy (Table 7 and Table 8) were fairly similar to those noted earlier for the entire experiment and(or) during the time of feces and urine collection. Minor differences would be due to the consideration of different days most appropriate for the specific measures and periods of time.

The ruminal methane emission in MJ/day was greater for KAT than for ALP (*p* < 0.05), but values relative to DM intake, BW^0.75^, and intakes of GE and DE did not differ (Table 7 and Table 8). Diet did not affect methane emission regardless of the expression. There were differences among periods for all expressions of methane emission, with the highest values for period 3 (*p* < 0.05) except for the expression of kJ/kg BW^0.75^. Likewise, values were lowest (*p* < 0.05) for period 1 except for kJ/kg BW^0.75^. For that expression, the value was lowest for period 4 (*p* < 0.05). The values for the intakes of GE, DE, and ME in MJ/day were approximately 30% greater for KAT than for ALP, and HE for KAT was approximately 28% greater than for ALP as well (*p* < 0.05). However, HE in kJ/kg BW^0.75^ was similar between breeds. Urinary energy decreased with increasing dietary concentration of lespedeza (*p* < 0.05), although ME intake was similar among diets. HE in MJ/day was similar among diets, but expressed as kJ/kg BW^0.75^ it was greatest among diets for ALF+LES. There was an interaction between breed and period in HE expressed as MJ/day (*p* < 0.05), with greater differences among periods for KAT vs. ALP. There was an interaction between diet and period in HE expressed as kJ/kg BW^0.75^ (*p* < 0.05), with generally greater and more consistent decreases with an advancing period for ALF and ALF+LES compared with LES. RE in MJ/day was greater for KAT vs. ALP (*p* < 0.05), but values relative to intakes of GE, DE, and ME were similar between breeds. There were no effects of diet on any expression of RE. The period affected RE regardless of the expression, with the greatest values among periods for period 1 (*p* < 0.05).

### 3.5. Ruminal Fluid and Plasma Measures

As noted earlier, there were three-way interactions (*p* < 0.05) for a number of ruminal fluid variables, which were concentrations of acetate, propionate, butyrate, isovalerate, and valerate, the acetate to propionate ratio, and the number of bacteria (Table 9 and Table 10). Likewise, there were many significant two-way interactions between breed and diet with the analysis conducted by period. As a result, it would be quite difficult to clearly describe these findings to facilitate a meaningful interpretation in line with objectives of the study based on the analysis by period. Hence, the main effect and two-way interaction means will receive attention, although means for the analysis by period are presented as well for completeness and potential future use for purposes such as meta-analyses.

The ruminal pH was greater for KAT vs. ALP, the greatest among diets for LES, and ranked period 2 < 3 and 4 < 1 (*p* < 0.05; Table 9 and Table 10). The concentration of AMN was greater for ALP vs. KAT, the lowest among diets for LES, and greater in period 3 than 2 (*p* < 0.05). The concentration of total VFA was greater for ALP than for KAT, the lowest among diets for LES, and the lowest among periods for period 3 (*p* < 0.05).

The molar percentage of acetate was greater and that of propionate lower for KAT than for ALP, and there was a corresponding difference in the ratio of acetate to propionate (*p* < 0.05; Table 9 and Table 10). The molar percentage of acetate was the lowest among diets for ALF and that of propionate was the greatest for ALF, also with a corresponding difference in acetate:propionate (*p* < 0.05). There were differences among periods in molar proportions of acetate and propionate as well. The molar proportion of acetate was lower and that of propionate greater in period 1 than in periods 3 and 4 (*p* < 0.05). The molar percentages of isobutyrate and isovalerate were greater for ALP vs. KAT, greater for ALF than for diets with lespedeza, and greater in periods 3 and 4 than in periods 1 and 2 (*p* < 0.05). The concentration of butyrate was greater for ALP vs. KAT, greatest among diets for LES, and the lowest among periods for period 4 (*p* < 0.05). The molar proportion of valerate was greater for ALP vs. KAT, ranked ALF > ALF+LES > LES, and was greatest among periods for period 3 (*p* < 0.05).

The numbers of bacteria and protozoa in ruminal fluid were greater for ALP than for KAT (*p* < 0.05; Table 9 and Table 10). The number of bacteria was lowest among diets for LES, and the number of protozoa was lower for ALF+LES and LES than for ALF (*p* < 0.05). The number of bacteria ranked period 1 > 2 > 3 and 4, and the number of protozoa was lowest among periods for period 1 (*p* < 0.05).

The concentration of UN in plasma was greater for KAT than for ALP, ranked ALF > ALF+LES > LES, and ranked period 1 and 4 < 2 < 3 (*p* < 0.05; Table 9 and Table 10). However, the interaction between breed and diet was due to similar values with LES compared with greater means for KAT with ALF and ALF+LES. The TAC of plasma did not differ between breeds, and the main effect of diet was not significant (*p* > 0.05). However, there was an effect of period and a diet × period interaction (*p* < 0.05). TAC was similar among the diets in period 1, greatest among the diets for ALF+LES in period 2, lowest among the diets for ALF in period 3, and greater for LES than for ALF+LES in period 4. Moreover, for ALF and LES, TAC was the greatest among periods in period 4, whereas the greatest value for ALF+LES was in period 2.

## 4. Discussion

### 4.1. Feed Intake and Growth Performance

Generally, diets and forages containing high concentrations of tannins reduce feed intake; however, this can vary with factors such as animal species and the type of tannins [18,41]. It has been suggested that goats are relatively tolerant to CT-containing forages with concentrations up to 10%, with feed intake by sheep reduced at lower concentrations [18]. In the present study, total feed intake by KAT was not affected by diet, suggesting that this breed of hair sheep is relatively tolerant of moderate to high dietary concentrations of CT.

The overall lack of effect of dietary concentration of lespedeza on feed intake is in accordance with the results of some but not all previous studies with goats and sheep at this institution. For example, the dietary inclusion of Kobe lespedeza (*Lespedeza striata*) containing 15.1% CT at between 0 and 100%, replacing sorghum–sudangrass, did not affect the feed intake by meat goats [10]. Similarly, intake by mature Boer goats of pelleted Sericea lespedeza with 6.4% CT and supplemented with 0.5% BW of rolled corn was not different from the intake of pelleted alfalfa [38]. Conversely, the intake of a 25% concentrate, 75% forage diet, with forage of Sericea lespedeza, plus the addition of quebracho extract to achieve a dietary CT concentration of 8.4%, was 16% less than with alfalfa hay as the basal forage [39]. In addition, there are reports of an increased intake of diets containing lespedeza, with the intake by Katahdin and St. Croix sheep ewes of a diet of Sericea lespedeza hay (5.8% CT) 28% greater than of alfalfa hay [14]. Turner et al. [42] also noted a greater total feed intake by Boer × Spanish goats of Sericea lespedeza than alfalfa hay (i.e., 4.21 vs. 3.87% BW). Based on these reports, it may not be surprising that the dietary inclusion of Sericea lespedeza did not have a consistent effect on feed intake by small ruminants in the meta-analysis study of Pech-Cervantes et al. [1].

Although the feed intake was not affected by diet, ADG decreased as the dietary amount of lespedeza increased, being 27 and 48 g greater for ALF than for ALF+LES and LES, respectively. In accordance, ADG:DM intake was affected by diet, although values for ALF+LES and LES were similar. Reduced ADG in small ruminants fed Sericea lespedeza forage has been reported in a number of studies (e.g., [1,38,39,43]). Decreased ADG is usually associated with reduced digestibility, and in this case presumably also with the lower CP content of diets with lespedeza, in particular LES. Similar RE among the diets supports a major impact of dietary N in this experiment.Conversely, in some studies in which ADG by goats was not affected [44,45,46], or even with increased ADG [47], diets with lespedeza had similar or greater concentrations of CP as the control diet. In support of the potential impact of limited N and amino acid absorption on growth performance, digested CP was 6.7% of the DM intake for LES, with values of 11.5 and 17.3% for ALF+LES and ALF, respectively. By applying 88% true protein digestibility and 2.67% metabolic fecal CP for goats determined by Moore et al. [48], a dietary CP concentration of 10.6% is predicted for LES.

### 4.2. Digestibility

The magnitude of the impact on digestibility of the dietary concentration of lespedeza ranked N > NDF > OM. In part, this would suggest little to no effect on digestion of non-forage constituents of the diet, mainly rolled corn and molasses. The reduced NDF digestibility of diets with lespedeza can be attributed to the relatively greater lignin content compared with this source of alfalfa. Lignin is responsible for the reduced digestibility of forages due to bonding with other fiber fractions as well as providing a physical barrier (i.e., encrusting) that retards or prevents microbial access to potentially degradable cellulose and hemicellulose [49]. Furthermore, diets rich in CT have antimicrobial properties that can inhibit the growth and activity of fibrolytic bacteria and(or) reduce enzyme activity [18,50,51]. The reduced DM and fiber digestibility of diets containing lespedeza have been reported in many studies, particularly with high dietary concentrations [3,10,13,37,39].

Reduced total tract N digestibility with the lespedeza diets can be attributed to the formation of CT–protein complexes that are not degraded in the intestine, a greater proportion of neutral detergent or acid detergent insoluble N, and(or) inhibition of protein-degrading bacteria and proteolytic enzymes in the rumen by CT. In support, CT of *Lotus corniculatus* and *Lotus pedunculatus* inhibited the growth of select protein-degrading bacteria and lowered the proteolytic enzyme activity [51,52,53]. Ideally, tannin–protein complexes formed in the rumen should be soluble in the acidic environment of the abomasum, with protein being dissociated from complexes to allow enzymatic digestion in the intestine and subsequent amino acid absorption [41,54]. However, insoluble CT–protein complexes that do not dissociate may be formed when the ratio of CT to total protein is relatively high [55]. Because LES had a relatively low concentration of CP and a high concentration of CT, it is likely that an appreciable quantity of CT–protein complexes formed in the rumen did not dissociate in the intestine, resulting in a substantial decrease in total tract N digestion. Moreover, lespedeza used by Terrill et al. [2] contained 0.9 to 1.1% of neutral detergent insoluble N (i.e., 31 to 46% of total CP) and 0.5 to 0.7% of acid detergent insoluble N (i.e., 17 to 23% of total CP), with these fractions being of low digestibility or indigestible. The impact of the amount of lespedeza in the diet on N digestion was indeed marked, with predicted digestibility of 75.4, 72.4, and 67.5% for ALF, ALF+LES, and LES, respectively, based on assumptions of true protein digestion and metabolic fecal CP of Moore et al. [48] noted earlier.

The breed × diet interaction in N digestion, due to the lower digestibility for KAT vs. ALP with LES but not other diets, indicates that CT exerted a greater adverse effect on N digestion in sheep compared with goats, which has also been reported by Min and Solaiman [18]. The lesser effect of CT on N digestion in goats might be attributed to a greater ability of microbiota (e.g., *Streptococcus caprinus*) to degrade tannins, a greater capacity for urea recycling, and(or) a greater quantity of saliva secreted or different salivary protein characteristics [18,56]. The decrease in UN excretion as the dietary concentration of lespedeza increased would be a direct consequence of decreased N absorption. Diets supplemented with sources of CT or fed CT-containing forage usually cause a shift in N excretion from urine to feces, which is beneficial environmentally for reducing N pollution [57,58,59].

### 4.3. Methane Emission and Energy Metabolism

The dietary inclusion of lespedeza has decreased methane emissions in many studies, presumably because of the presence of CT [1,3,10,12,13,37,39]. CT have direct effects on methane production by inhibiting the activity and growth of methanogenic bacteria, and indirect effects can occur as well as through the reduced digestibility of OM, especially of fiber, thereby decreasing the availability of hydrogen for use in methanogenesis [7,41]. CT present in lespedeza can also lessen the number of protozoa, which can lead to decreased methane production because of symbiotic associations between protozoa and methanogens [60,61]. Based on these conditions, it is perhaps somewhat surprising that the dietary inclusion of lespedeza did not affect methane production in this study. Methanogenic bacteria were not characterized, although the number of protozoa was lower for diets containing lespedeza. Sometimes, methanogenesis not related to the presence or actions of protozoa may increase when protozoa are inhibited in response to changes in the diversity and(or) the abundance of free-living methanogens not physically associated with protozoa [8,62].

The concentration and nature of fiber and protein in the diet can influence ruminal methane production [7]. In the present study, alfalfa was of high quality, with the concentration of fiber fractions much greater and that of protein lower in lespedeza. Diets rich in fiber promote high numbers and activity of ruminal fibrolytic microbiota that produce acetate and hydrogen, possibly stimulating methanogenesis [7,63]. In accordance, the molar proportion of acetate increased with increasing dietary concentrations of lespedeza. Overall, these factors could have contributed to no diet effect on methane production. van Cleef et al. [15] recently noted a similar lack of effect of dietary concentration of Sericea lespedeza hay and bermudagrass hay (i.e., 0, 25, 50, 75, and 100%) on ruminal methane emission by beef steers, in g/day, g/kg OM intake, and g/kg digested OM intake. Although, an effect on methane emission relative to CT intake was noted [15]. As in the present experiment, digestibilities decreased as the concentration of lespedeza increased. However, this would not appear to have involved the dietary concentration of CP. Because of the use of bermudagrass hay as the other forage, the dietary concentration of CP markedly increased and that of NDF decreased as the concentration of lespedeza increased, in contrast to the current experiment.

The decrease in urinary energy with increasing dietary lespedeza amounts was in accordance with most energy in urine primarily from nitrogenous compounds. It was somewhat unexpected that the ruminal AMN concentration was similar between ALF and ALF+LES, given that urinary N and energy and plasma UN concentration ranked ALF > ALF+LES > LES. Although ME intake and HE in MJ/day were similar among diets, HE in kJ/kg BW^0.75^ was slightly greater for ALF+LES than for ALF (5.2%) and LES (7.1%). This was not expected based on ADG as well, although it may be germane to note that kJ/kgBW^0.75^ values are based on BW determined at the beginning and end of calorimetry measures. There was no difference in HE between breeds and species in this study, which was also observed by El-Meccawi et al. [64]. Energy retention expressed as a percentage of GE, DE, or ME intake did not differ among diets or between goats and sheep. This suggests that the effects of lespedeza and its CT were restricted mainly to digestion rather than the efficiency of energy metabolism. Relatedly, as alluded to earlier, limited total tract N digestion and intestinal amino acid absorption may have been largely responsible for decreasing ADG as the dietary amounts of lespedeza increased. Furthermore, it can be postulated that if the diets had been formulated differently, such as the inclusion of a high-protein feedstuff, for example, soybean meal in diets with lespedeza rather than corn, perhaps feed intake would have been greater as has been observed in some studies. One could even hypothesize that with such different conditions, perhaps diet effects on variables such as ruminal methane emission would have differed from those observed.

### 4.4. Ruminal Conditions

The highest ruminal pH among diets for LES is in agreement with the lowest concentration of VFA in ruminal fluid, although DE intake was similar among diets. A higher ruminal pH for KAT than for ALP also was associated with a lower concentration of total VFA. As noted before, the relatively low ruminal AMN concentration for LES, at least in part, relates to the binding of CT to protein, preventing degradation in the rumen [41]. Another reason for decreased AMN concentrations for the diets with lespedeza might be the inhibitory effects of CT on protozoa [60,61]. Protozoa are responsible for bacterial protein turnover and dietary protein degradation, resulting in the decreased efficiency of protein utilization and increased AMN concentrations [61,65]. Min and Solaiman [18] reported that CT in forage decreases protein degradation to a greater extent in sheep than goats; however, a breed × diet interaction was not observed in this study.

As noted before, three-way interactions in variables such as molar percentages of acetate and propionate make clear interpretation difficult. In regard to the overall greater molar percentage of acetate and the lower proportion of propionate for KAT than for ALP, contrasting findings in the literature exist. For example, a greater proportion of acetate was noted in sheep by Ramos-Morales et al. [66], Isac et al. [67] reported a greater proportion in goats, and no species difference was observed by Hadjipanayiotou and Antoniou [68] or Yanez Ruiz et al. [69]. Decreasing molar proportions of isobutyrate, isovalerate, and valerate with increasing concentrations of lespedeza in the diet are probably because branched-chain VFA (i.e., isobutyrate, isovalerate, and 2-methylbutyrate) are produced from deamination and decarboxylation of branched-chain amino acids (i.e., valine, leucine, and isoleucine, respectively), along with valerate (from amino acids and carbohydrate fermentation) by ruminal microorganisms [70], again with CT decreasing the ruminal degradation of protein. Similar to findings of this study, Ramos-Morales et al. [66] noted a greater number of protozoa in goats than sheep.

## 5. Conclusions

The major effect of the dietary amounts of lespedeza was on digestibility, with the magnitude of change ranking N > NDF > OM. The total tract N digestion in goats with the diet highest in lespedeza was affected less than in sheep, suggesting some means of greater tolerance to impacts of CT in goats. Decreasing total tract N digestion and, thus, intestinal amino acid absorption with increasing dietary lespedeza, appeared responsible for decreasing ADG, without an apparent impact on energy absorption or efficiency of utilization. The dietary concentration of lespedeza and CT did not affect ruminal methane emissions with goats or sheep regardless of the length of feeding, in contrast to many other experiments. Further research should address such effects with different diet formulations to lessen the potential impact of the effects of CT on intestinal amino acid absorption and other conditions that it could impact.

## Figures and Tables

**Table 1 animals-12-02064-t001:** Ingredient and chemical composition of diets fed to Alpine doelings and Katahdin ewe lambs.

	Diet ^1^		
Item	ALF	ALF+LES	LES
Ingredient (% dry matter)			
Alfalfa hay	75.00	37.50	0.00
Lespedeza hay	0.00	37.50	75.00
Rolled corn	19.32	19.32	19.32
Molasses	5.00	5.00	5.00
Dicalcium phosphate	0.08	0.08	0.08
Mineral supplement ^2^	0.50	0.50	0.50
Vitamin premix ^3^	0.05	0.05	0.05
Trace mineral mix ^4^	0.05	0.05	0.05
Constituent ^5^ (% dry matter)			
Ash	10.5 ± 0.29	8.9 ± 0.23	7.4 ± 0.30
Crude protein	21.2 ± 0.63	17.1 ± 0.42	13.0 ± 0.27
Neutral detergent fiber	35.3 ± 0.70	39.0 ± 0.58	40.0 ± 1.06
Acid detergent fiber	25.1 ± 0.67	31.1 ± 0.46	32.7 ± 1.04
Acid detergent lignin	6.9 ± 0.24	10.1 ± 0.32	11.3 ± 0.34
Condensed tannins	0.9 ± 0.07	5.8 ± 0.48	10.0 ± 0.94

^1^ ALF = alfalfa; ALF+LES = alfalfa and lespedeza in 1:1 ratio; LES = lespedeza. ^2^ 9–10% Ca, 6% P, 35–40% NaCl, 1% Mg, 1% K, 1% S, 125 mg/kg Co, 150 mg/kg I, 5000 mg/kg Fe, 10 mg/kg Se, 140 mg/kg Zn, 352,000 IU/kg vitamin A, 88,000 IU/kg vitamin D_3_, and 330 IU/kg vitamin E; Stillwater Milling, Stillwater, OK, USA. ^3^ 8,800,000 IU/kg vitamin A, 1,760,000 IU/kg vitamin D_3_, and 1100 IU/kg vitamin E; NB-8006, Nutra Blend, Neosho, MO, USA. ^4^ 275 mg/kg Co, 2000 mg/kg I, 43,746 mg/kg Fe, 750 mg/kg Se, 18,748 mg/kg Cu, 68,744 mg/kg Zn, and 19,998 mg/kg Mn. ^5^ SEM based on weekly samples.

**Table 2 animals-12-02064-t002:** Average daily temperature (T), relative humidity (RH), and temperature–humidity index (THI) in the facility in which animals were housed ^1^.

Period	Item	Mean	SEM	Minimum	Maximum
1	Temperature (°C)	13.6	0.08	4.1	21.9
	Relative humidity (%)	53.4	0.22	29.3	74.7
	THI	56.8	0.11	44.1	67.8
2	Temperature (°C)	14.9	0.12	−0.9	30.6
	Relative humidity (%)	60.5	0.37	19.7	99.0
	THI	58.2	0.16	33.8	75.6
3	Temperature (°C)	20.2	0.12	7.5	32.2
	Relative humidity (%)	73.9	0.37	28.4	98.9
	THI	66.4	0.16	46.4	80.3
4	Temperature (°C)	26.8	0.10	15.2	38.2
	Relative humidity (%)	70.3	0.34	30.8	97.2
	THI	76.0	0.11	59.2	86.5

^1^ THI = (0.8 × T) + (RH × ((T − 14.3)/100)) + 46.3 [40].

**Table 3 animals-12-02064-t003:** *p* values for effects of breed, diet, and period on body weight, dry matter intake, average daily gain, gain efficiency, and the Kleiber ratio during the entire experiment ^1^.

	Source of Variation						
Item ^2^	Breed	Diet	Breed × Diet	Period	Breed × Period	Diet × Period	Breed × Diet × Period
BW (kg)	<0.001	0.183	0.499	<0.001	<0.001	<0.001	0.582
DMI							
g/day	<0.001	0.199	0.346	<0.001	0.002	0.486	0.211
% BW	0.063	0.540	0.692	<0.001	0.535	0.132	0.067
g/kg BW^0.75^	0.001	0.425	0.566	<0.001	0.960	0.229	0.077
ADG (g)	<0.001	<0.001	0.326	<0.001	0.063	0.013	0.117
ADG:DMI (g/kg)	<0.001	0.001	0.326	<0.001	0.018	0.108	0.479
KR (g/kg BW^0.75^)	<0.001	<0.001	0.691				
RFI, breed (g/day)	1.000	0.432	0.713				
RFI, combined (g/day)	0.851	0.337	0.540				

^1^ Breeds were Alpine doelings and Katahdin ewe lambs; diets were 75% forage consisting of alfalfa hay, a 1:1 mixture of alfalfa and Sericea lespedeza hay, and lespedeza hay; periods 1–3 were 42 days in length and period 4 was 47 days. ^2^ BW = body weight; DMI = dry matter intake; ADG = average daily gain; KR = Kleiber ratio; RFI = residual feed intake; RFI, breed = based on separate equations for each breed; RFI, combined = based on one equation for both breeds.

**Table 4 animals-12-02064-t004:** Effects of breed, diet, and period on dry matter intake, average daily gain, gain efficiency, and the Kleiber ratio during the entire experiment^1^.

	Interaction		Breed			Diet				Period				
Item ^2^	Breed	Diet	ALP	KAT	SEM	ALF	ALF+LES	LES	SEM	1	2	3	4	SEM
BW (kg)			33.5 ^a^	44.4 ^b^	1.04	40.9	39.2	37.5	1.28	30.0 ^a^	36.6 ^b^	42.6 ^c^	47.5 ^d^	0.75
	ALP									27.3 ^a^	31.7 ^b^	36.0 ^c^	39.1 ^d^	1.06
	KAT									32.7 ^b^	41.5 ^d^	49.2 ^e^	55.9 ^f^	
		ALF								30.0 ^a^	37.8 ^bc^	44.9 ^ef^	50.9 ^g^	1.30
		ALF-LES								29.9 ^a^	36.6 ^b^	42.7 ^de^	47.4 ^fg^	
		LES								30.2 ^a^	35.5 ^b^	40.1 ^cd^	44.3 ^ef^	
DMI														
g/day			1274 ^a^	1817 ^b^	51.1	1600	1584	1452	62.6	1242 ^a^	1635 ^b^	1682 ^b^	1623 ^b^	43.9
	ALP									1063 ^a^	1361 ^b^	1372 ^b^	1300 ^b^	62.0
	KAT									1420 ^b^	1909 ^c^	1992 ^c^	1946 ^c^	
% BW			3.84	4.14	0.110	3.97	4.10	3.89	0.134	4.14 ^b^	4.45 ^c^	3.94 ^b^	3.43 ^a^	0.102
g/kg BW^0.75^			91.9 ^a^	105.9 ^b^	2.60	99.3	101.7	95.8	3.18	95.6 ^b^	109.2 ^c^	100.3 ^b^	89.6 ^a^	2.45
ADG (g)			88 ^a^	180 ^b^	5.0	159 ^c^	132 ^b^	111 ^a^	6.1					
		ALF								167 ^e^	204 ^f^	137 ^cde^	129 ^cd^	12.3
		ALF+LES								147 ^de^	169 ^e^	125 ^cd^	88 ^ab^	
		LES								149 ^de^	103 ^abc^	116 ^bcd^	76 ^a^	
ADG:DMI (g/kg)			72 ^a^	104 ^b^	3.4	101 ^b^	84 ^a^	79 ^a^	4.2					
	ALP									93 ^cd^	84 ^bc^	73 ^b^	38 ^a^	6.8
	KAT									147 ^e^	108 ^d^	86 ^bc^	75 ^bc^	
KR (g/kg BW^0.75^)			6.33 ^a^	10.46 ^b^	0.185	9.70 ^c^	8.32 ^b^	7.15 ^a^	0.226					

^1^ Breeds were Alpine doelings (ALP) and Katahdin ewe lambs (KAT); diets were 75% forage consisting of alfalfa hay (ALF), a 1:1 mixture of alfalfa and Sericea lespedeza hay (ALF+LES), and lespedeza hay (LES); periods 1–3 were 42 days in length and period 4 was 47 days. ^2^ BW = body weight; DMI = dry matter intake; ADG = average daily gain; KR = Kleiber ratio; RFI = residual feed intake; RFI, breed = based on separate equations for each breed; RFI, combined = based on one equation for both breeds. ^a,b,c,d,e,f,g^ Means within grouping without a common superscript letter differ (*p* < 0.05).

**Table 5 animals-12-02064-t005:** *p* values for effects of breed, diet, and period on intake and digestion during feces and urine collection ^1^.

	Source of Variation						
Item ^2^	Breed	Diet	Breed × Diet	Period	Breed × Period	Diet × Period	Breed × Diet × Period
Dry matter							
Intake							
g/day	<0.001	0.577	0.991	0.034	0.012	0.838	0.408
% BW	0.901	0.120	0.716	<0.001	0.390	0.186	0.555
g/kg BW^0.75^	0.118	0.122	0.778	<0.001	0.331	0.525	0.413
Digestion (%)	0.483	<0.001	0.213	<0.001	0.862	0.735	0.314
Digested (g/day)	<0.001	0.255	0.691	0.601	0.072	0.886	0.400
Organic matter							
Intake (g/day)	<0.001	0.319	0.993	0.060	0.013	0.805	0.403
Digestion (%)	0.704	<0.001	0.227	<0.001	0.797	0.637	0.317
Digested (g/day)	<0.001	0.531	0.751	0.589	0.054	0.885	0.413
Neutral detergent fiber							
Intake (g/day)	<0.001	0.007	0.952	0.065	0.011	0.143	0.362
Digestion (%)	0.839	<0.001	0.130	<0.001	0.781	0.053	0.361
Digested (g/day)	<0.001	0.017	0.273	0.716	0.153	0.107	0.429
Nitrogen							
Intake (g/day)	<0.001	<0.001	0.433	<0.001	0.008	0.168	0.466
Digestion (%)	0.037	<0.001	0.009	0.154	0.491	0.190	0.373
Digested (g/day)	<0.001	<0.001	0.047	<0.001	0.027	0.085	0.463
Urine (g/day)	0.009	<0.001	0.191	<0.001	<0.001	0.044	0.405
Retained (g/day)	0.001	<0.001	0.302	0.013	0.842	0.061	0.879
DE intake (MJ/day)	<0.001	0.531	0.751	0.589	0.054	0.885	0.413
Urine energy (MJ/day)	0.013	<0.001	0.272	<0.001	0.013	0.701	0.506

^1^ Breeds were Alpine doelings and Katahdin ewe lambs; diets were 75% forage consisting of alfalfa hay, a 1:1 mixture of alfalfa and Sericea lespedeza hay, and lespedeza hay; periods 1–3 were 42 days in length and period 4 was 47 days; feces and urine were collected for 5 days in the last week of the periods; intake was the average of values 2 days before the first 4 days of collection. ^2^ BW = body weight; DE = digestible energy.

**Table 6 animals-12-02064-t006:** Effects of breed, diet, and period on intake and digestion during feces and urine collection ^1^.

	Interaction		Breed			Diet				Period				
Item ^2^	Breed	Diet	ALP	KAT	SEM	ALF	ALF+LES	LES	SEM	1	2	3	4	SEM
Dry matter														
Intake														
g/day			1252 ^a^	1660 ^b^	45.5	1411	1492	1465	55.7					
	ALP									1250 ^a^	1314 ^a^	1262 ^a^	1182 ^a^	61.4
	KAT									1523 ^b^	1633 ^bc^	1789 ^d^	1697 ^c^	
% BW			3.83	3.86	0.115	3.60	3.96	3.97	0.141	4.65 ^d^	4.06 ^c^	3.60 ^b^	3.07 ^a^	0.110
g/kg BW^0.75^			89.0	94.6	2.50	86.5	94.3	94.6	3.07	102.5 ^c^	96.2 ^c^	88.8 ^b^	79.8 ^a^	2.54
Digestion (%)			68.7	68.0	0.73	73.7 ^c^	67.5 ^b^	63.7 ^a^	0.89	71.9 ^b^	67.2 ^a^	67.0 ^a^	67.2 ^a^	0.79
Digested (g/day)			867 ^a^	1136 ^b^	37.1	1048	1015	942	45.4	1000	997	1034	976	37.0
Organic matter														
Intake (g/day)			1140 ^a^	1511 ^b^	41.4	1262	1359	1356	50.8	1270	1343	1384	1306	39.6
	ALP									1145 ^ab^	1199 ^b^	1145 ^ab^	1073 ^a^	56.0
	KAT									1395 ^c^	1488 ^cd^	1623 ^e^	1539 ^de^	
Digestion (%)			70.2	69.9	0.70	75.3 ^c^	69.3 ^b^	65.5 ^a^	0.86	73.3 ^b^	68.7 ^a^	69.0 ^a^	69.2 ^a^	0.75
Digested (g/day)			807 ^a^	1062 ^b^	34.2	958	949	896	41.9	933	929	965	912	34.0
NDF														
Intake (g/day)			478 ^a^	633 ^b^	17.4	497 ^a^	582 ^b^	586 ^b^	21.3	528	565	574	554	16.5
	ALP									476 ^a^	505 ^a^	474 ^a^	456 ^a^	23.3
	KAT									580 ^b^	625 ^bc^	674 ^c^	652 ^c^	
Digestion (%)			51.4	51.0	1.22	61.7 ^c^	50.5 ^b^	41.4 ^a^	1.49	56.6 ^b^	49.1 ^a^	48.7 ^a^	50.6 ^a^	1.32
Digested (g/day)			249 ^a^	325 ^b^	12.8	312 ^b^	299 ^b^	249 ^a^	15.6	297	280	284	285	13.2
Nitrogen														
Intake (g/day)			34.1 ^a^	45.4 ^b^	1.25	47.9 ^c^	40.9 ^b^	30.4 ^a^	1.53	35.1 ^a^	40.6 ^b^	43.6 ^c^	39.8 ^b^	1.21
	ALP									31.6 ^a^	36.0 ^bc^	36.2 ^bc^	32.7 ^ab^	1.71
	KAT									38.5 ^c^	45.2 ^d^	51.0 ^e^	46.9 ^d^	
Digestion (%)			66.6 ^b^	64.3 ^a^	0.75	78.8 ^c^	66.9 ^b^	50.8 ^a^	0.92	66.0	64.2	66.6	65.2	0.87
	ALP					78.5 ^d^	67.0 ^c^	54.4 ^b^	1.30					
	KAT					79.0 ^d^	66.7 ^c^	47.3 ^a^						
Digested (g/day)			23.5 ^a^	30.6 ^b^	1.03	38.0 ^c^	27.5 ^b^	15.7 ^a^	1.26	23.9 ^a^	27.2 ^b^	30.3 ^c^	26.9 ^b^	1.04
	ALP					32.3 ^c^	23.7 ^b^	14.5 ^a^	1.78	22.1 ^a^	24.3 ^a^	25.4 ^a^	22.2 ^a^	1.47
	KAT					43.7 ^d^	31.3 ^c^	16.8 ^a^		25.6 ^a^	30.1 ^b^	35.1 ^c^	31.7 ^b^	
Urine (g/day)			12.4 ^a^	14.7 ^b^	0.60	19.4 ^c^	14.0 ^b^	7.3 ^a^	0.74	9.2 ^a^	14.6 ^b^	15.1 ^b^	15.4 ^b^	0.57
	ALP									9.5 ^a^	13.7 ^bc^	12.9 ^b^	13.6 ^bc^	0.80
	KAT									8.9 ^a^	15.5 ^c^	17.3 ^d^	17.2 ^d^	
		ALF								13.8 ^c^	20.8 ^d^	20.6 ^d^	22.6 ^d^	0.98
		ALF+LES								9.4 ^b^	14.8 ^c^	16.5^c^	15.0 ^c^	
		LES								4.3 ^a^	8.2 ^b^	8.1 ^b^	8.8 ^b^	
Retained (g/day)			11.1 ^a^	15.9 ^b^	0.97	18.6 ^c^	13.6 ^b^	8.3 ^a^	1.19	14.7 ^bc^	12.6 ^ab^	15.2 ^c^	11.5 ^a^	1.03
DE intake (MJ/day)			15.59 ^a^	20.53 ^b^	0.661	18.52	18.35	17.32	0.810	18.03	17.95	18.64	17.62	0.657
UE (MJ/day)			0.59 ^a^	0.68 ^b^	0.025	0.81 ^c^	0.66 ^b^	0.43 ^a^	0.031	0.49 ^a^	0.66 ^b^	0.70 ^b^	0.69 ^b^	0.025
	ALP									0.49 ^a^	0.63 ^b^	0.60 ^b^	0.63 ^b^	0.035
	KAT									0.49 ^a^	0.69 ^bc^	0.79 ^d^	0.74 ^cd^	

^1^ Breeds were Alpine doelings (ALP) and Katahdin ewe lambs (KAT); diets were 75% forage consisting of alfalfa hay (ALF), a 1:1 mixture of alfalfa and Sericea lespedeza hay (ALF+LES), and lespedeza hay (LES); periods 1–3 were 42 days in length and period 4 was 47 days; feces and urine werecollected for 5 days in the last week of the periods; intake was the average of values 2 days before the first 4 days of collection. ^2^ BW = body weight; NDF = neutral detergent fiber; DE = digestible energy; UE = urine energy. ^a,b,c,d,e^ Means without a grouping without a common superscript letter differ (*p* < 0.05).

**Table 7 animals-12-02064-t007:** *p* values for effects of breed, diet, and period on dry matter intake, heat energy, and ruminal methane emission during calorimetry system measures ^1^.

	Source of Variation						
Item ^2^	Breed	Diet	Breed × Diet	Period	Breed × Period	Diet × Period	Breed × Diet × Period
DM intake ^3^							
g/day	<0.001	0.541	0.920	0.210	0.185	0.033	0.392
% BW	0.552	0.043	0.880	<0.001	0.165	0.001	0.527
g/kg BW^0.75^	0.322	0.074	0.895	<0.001	0.203	0.002	0.475
GE intake (MJ/day)	<0.001	0.274	0.933	0.224	0.086	0.028	0.381
DE intake (MJ/day)	<0.001	0.845	0.659	0.085	0.184	0.112	0.392
UE (MJ/day)	0.083	<0.001	0.355	<0.001	0.197	0.565	0.720
Methane	0.001	0.487	0.968	<0.001	0.415	0.774	0.490
MJ/day	0.001	0.487	0.968	<0.001	0.415	0.774	0.490
kJ/g DM intake	0.228	0.742	0.462	<0.001	0.574	0.744	0.462
kJ/kg BW^0.75^	0.658	0.132	0.946	<0.001	0.929	0.536	0.508
% GE intake	0.226	0.547	0.464	<0.001	0.578	0.716	0.464
% DE intake	0.418	0.351	0.340	<0.001	0.800	0.873	0.311
ME intake (MJ/day)	<0.001	0.942	0.658	0.052	0.209	0.124	0.394
Heat energy							
MJ/day	<0.001	0.064	0.234	<0.001	<0.001	0.691	0.800
kJ/kg BW^0.75^	0.128	0.027	0.323	<0.001	0.426	0.028	0.796
Retained energy							
MJ/day	0.016	0.870	0.820	0.006	0.330	0.183	0.361
% GE intake	0.943	0.489	0.486	<0.001	0.826	0.364	0.197
% DE intake	0.881	0.628	0.346	0.001	0.917	0.378	0.278
% ME intake	0.989	0.687	0.276	0.004	0.941	0.353	0.242

^1^ Breeds were Alpine doelings and Katahdin ewe lambs; diets were 75% forage consisting of alfalfa hay, a 1:1 mixture of alfalfa and Sericea lespedeza hay, and lespedeza hay; periods 1–3 were 42 days in length and period 4 was 47 days; measures occurred in the last week of the periods. ^2^ DM = dry matter; BW = body weight; DE = digestible energy; UE = urine energy; ME = metabolizable energy; GE = gross energy. ^3^ Intake was based on 2 days prior to and the day of calorimetry measures.

**Table 8 animals-12-02064-t008:** Effects of breed, diet, and period on DM intake, heat energy, and ruminal methane emission during calorimetry system measures ^1^.

	Interaction		Breed			Diet				Period				
Item ^2^	Breed	Diet	ALP	KAT	SEM	ALF	ALF+LES	LES	SEM	1	2	3	4	SEM
DM intake ^3^														
g/day			1310 ^a^	1707 ^b^	54.0	1449	1530	1546	66.1					
		ALF								1434 ^a^	1456 ^ab^	1463 ^ab^	1443 ^ab^	89.4
		ALF+LES								1686 ^b^	1454 ^ab^	1582 ^ab^	1399 ^a^	
		LES								1415 ^a^	1502 ^ab^	1687 ^b^	1581 ^ab^	
% BW			3.90	3.80	0.121	3.55 ^a^	3.90 ^ab^	4.09 ^b^	0.148					
		ALF								4.44 ^e^	3.70 ^cd^	3.22 ^ab^	2.83 ^a^	0.204
		ALF+LES								5.18 ^f^	3.80 ^cd^	3.63 ^bcd^	3.00 ^a^	
		LES								4.42 ^e^	4.19 ^de^	4.17 ^de^	3.58 ^bc^	
g/kg BW^0.75^			93.6	97.8	2.95	89.1	97.2	100.0	3.62					
		ALF								105.6 ^e^	92.4 ^bcd^	83.2 ^ab^	75.4 ^a^	5.06
		ALF+LES								123.5 ^f^	94.2 ^bcde^	92.9 ^bcde^	78.1 ^a^	
		LES								104.9 ^de^	102.2 ^cde^	104.8 ^de^	91.8 ^abc^	
GEI (MJ/day)			23.06 ^a^	30.05 ^b^	0.948	25.06	26.94	27.66	1.16	26.77	25.92	27.68	25.85	0.908
		ALF								24.74 ^a^	25.35 ^a^	25.18 ^a^	24.97 ^a^	1.572
		ALF+LES								29.97 ^bc^	25.40 ^a^	27.86 ^abc^	24.54 ^a^	
		LES								25.59 ^ab^	27.02 ^abc^	29.99 ^c^	28.05 ^abc^	
UE (MJ/day)			0.62	0.69	0.031	0.83 ^c^	0.68 ^b^	0.46 ^a^	0.038	0.53 ^a^	0.67 ^b^	0.71 ^b^	0.70 ^b^	0.030
DEI (MJ/day)			16.28 ^a^	21.12 ^b^	0.760	18.99	18.84	18.27	0.931	19.67	17.90	19.25	17.98	0.746
Methane														
kJ/day			1175 ^a^	1436 ^b^	50.8	1248	1353	1316	62.2	1157 ^a^	1277 ^b^	1581 ^c^	1207 ^ab^	51.2
kJ/g DMI			0.918	0.865	0.0304	0.888	0.914	0.873	0.0372	0.776 ^a^	0.888 ^b^	1.044 ^c^	0.858 ^ab^	0.0370
kJ/kg BW^0.75^			83.5	81.7	2.83	76.8	85.4	85.7	3.46	85.6 ^b^	83.4 ^b^	94.1 ^b^	67.4 ^a^	2.99
% GEI			5.2	4.9	0.17	5.1	5.2	4.9	0.21	4.4 ^a^	5.0 ^b^	6.0 ^c^	4.9 ^ab^	0.21
% DEI			7.6	7.2	0.33	6.9	7.6	7.6	0.40	6.0 ^a^	7.5 ^b^	8.8 ^c^	7.3 ^b^	0.38
MEI (MJ/day)			14.52 ^a^	19.00 ^b^	0.728	16.94	16.82	16.52	0.891	18.03	15.96	16.97	16.08	0.724
Heat energy														
MJ/day			7.90 ^a^	10.19 ^b^	0.217	9.20	9.40	8.53	0.266					
	ALP									7.82 ^ab^	8.09 ^b^	8.16 ^b^	7.53 ^a^	0.249
	KAT									9.27 ^c^	10.20 ^d^	10.95 ^e^	10.33 ^d^	
kJ/kg BW^0.75^			560	579	8.4	563 ^a^	592 ^b^	553 ^a^	10.3					
		ALF								637 ^fg^	591 ^e^	548 ^cd^	476 ^a^	13.9
		ALF+LES								669 ^g^	608 ^ef^	585 ^de^	508 ^ab^	
		LES								590 ^de^	587 ^de^	540 ^bc^	496 ^a^	
RE														
MJ/day			6.62 ^a^	8.81 ^b^	0.616	7.74	7.42	7.99	0.754	9.49 ^b^	6.81 ^a^	7.41 ^a^	7.15 ^a^	0.666
% GEI			26.8	27.0	1.73	28.8	25.1	26.7	2.12	32.4 ^b^	24.5 ^a^	23.7 ^a^	25.1 ^a^	1.92
% DEI			37.0	37.5	2.47	37.4	35.1	39.2	3.02	46.3 ^b^	34.7 ^a^	33.1 ^a^	34.8 ^a^	2.81
% MEI			40.9	41.0	2.85	41.4	38.6	42.9	3.49	50.6 ^b^	38.5 ^a^	36.7 ^a^	38.1 ^a^	3.28

^1^ Breeds were Alpine doelings (ALP) and Katahdin ewe lambs (KAT); diets were 75% forage consisting of alfalfa hay (ALF), a 1:1 mixture of alfalfaand Sericea lespedeza hay (ALF+LES), and lespedeza hay (LES); periods 1–3 were 42 days in length and period 4 was 47 days. ^2^ GEI = gross energy intake; UE = urine energy; DEI = digestible energy intake; DMI = dry matter intake; BW = body weight; MEI = metabolizable energy intake; RE = retained energy. ^3^ Intake was based on 2 days prior to and the day of calorimetry measures. ^a,b,c,d,e,f,g^ Means within grouping without a common superscript letter differ (*p* < 0.05).

**Table 9 animals-12-02064-t009:** *p* values for effects of breed, diet, and period on ruminal fluid characteristics, numbers of bacteria and protozoa, and plasma constituent concentrations ^1^.

	Source of Variation							
Item ^2^	Period	Breed	Diet	Breed × Diet	Period	Breed × Period	Diet × Period	Breed × Diet × Period
Ruminal fluid								
pH		0.001	0.002	0.911	<0.001	0.191	0.463	0.838
AMN (mg/dL)		<0.001	<0.001	0.952	0.002	0.172	0.157	0.233
VFA								
Total (mmol/L)		0.028	<0.001	0.165	<0.001	0.575	0.003	0.369
Molar %								
Acetate		<0.001	<0.001	0.056	<0.001	<0.001	<0.001	0.001
	1	<0.001	<0.001	0.022				
	2	0.003	<0.001	0.254				
	3	0.011	0.003	0.402				
	4	0.724	<0.001	0.002				
Propionate		0.038	<0.001	0.039	0.002	0.004	0.041	0.026
	1	<0.001	<0.001	0.199				
	2	0.988	0.003	0.004				
	3	0.335	0.007	0.361				
	4	0.731	0.002	0.456				
Isobutyrate		0.016	<0.001	0.005	<0.001	0.005	<0.001	0.087
Butyrate		0.094	0.002	0.501	<0.001	0.456	0.306	0.003
	1	0.303	0.281	0.265				
	2	0.013	0.020	0.002				
	3	0.770	0.156	0.942				
	4	0.622	0.006	0.074				
Isovalerate		0.002	<0.001	0.004	0.003	0.002	<0.001	0.016
	1	0.003	0.043	0.290				
	2	<0.001	<0.001	0.083				
	3	<0.001	<0.001	0.815				
	4	0.334	<0.001	0.013				
Valerate		0.034	<0.001	0.010	0.001	<0.001	0.005	0.010
	1	0.055	<0.001	0.064				
	2	0.192	<0.001	0.165				
	3	0.001	<0.001	0.003				
	4	0.015	<0.001	0.065				
Acetate:propionate		0.002	<0.001	0.005	0.041	<0.001	0.068	0.015
	1	<0.001	<0.001	0.016				
	2	0.563	<0.001	0.004				
	3	0.155	0.001	0.343				
	4	0.517	<0.001	0.129				
Bacteria, ×10^10^/ML		0.005	0.002	0.276	<0.001	0.021	0.094	<0.001
	1	0.004	0.683	0.315				
	2	0.045	0.098	0.036				
	3	0.523	0.053	0.135				
	4	0.005	<0.001	<0.001				
Protozoa, ×10^5^/mL		0.030	<0.001	0.405	<0.001	0.317	0.067	0.468
Plasma								
UN (mg/L)		0.001	<0.001	0.010	<0.001	0.368	0.218	0.595
TAC (μmol/L)		0.077	0.152	0.051	<0.001	0.967	0.039	0.479

^1^ Breeds were Alpine doelings and Katahdin ewe lambs; diets were 75% forage consisting of alfalfa hay, a 1:1 mixture of alfalfa and Sericea lespedeza hay, and lespedeza hay; periods 1–3 were 42 days in length and period 4 was 47 days; samples were collected at 4 h after feeding in week 5 of each period. ^2^ AMN = ammonia nitrogen; VFA = volatile fatty acids; UN = urea nitrogen; TAC = total antioxidant capacity.

**Table 10 animals-12-02064-t010:** Effects of breed, diet, and period on ruminal ammonia and volatile fatty acid concentrations, numbers of bacteria and protozoa, and plasma constituent concentrations ^1^.

	Interaction ^2^			Breed			Diet				Period				
Item	Period	Diet	Breed	ALP	KAT	SEM	ALF	ALF+LES	LES	SEM	1	2	3	4	SEM
Ruminal fluid															
pH				6.01 ^a^	6.22 ^b^	0.040	6.00 ^a^	6.08 ^a^	6.27 ^b^	0.049	6.49 ^c^	5.81 ^a^	6.02 ^b^	6.14 ^b^	0.049
AMN (mg/dL)				10.1 ^b^	7.8 ^a^	0.42	10.4 ^b^	10.1 ^b^	6.4 ^a^	0.51	9.6 ^bc^	7.8 ^a^	9.9 ^c^	8.4 ^ab^	0.47
VFA															
Total (mmol/L)				72.9 ^b^	68.8 ^a^	1.28	78.4 ^b^	74.8 ^b^	59.4 ^a^	1.57	72.9 ^b^	74.5 ^b^	63.5 ^a^	72.6 ^b^	1.62
		ALF									86.4 ^f^	82.5 ^f^	66.2 ^bcd^	75.9 ^e^	2.83
		ALF+LES									75.2 ^de^	78.9 ^ef^	67.2 ^cd^	80.0 ^ef^	
		LES									59.3 ^ab^	59.4 ^abc^	57.1 ^a^	61.9 ^abc^	
Molar %															
Acetate				73.7 ^a^	75.2 ^b^	0.19	72.7 ^a^	75.0 ^b^	75.6 ^b^	0.24	73.5 ^a^	74.1 ^ab^	74.3 ^b^	76.0 ^c^	0.22
	1			71.9 ^a^	75.1 ^b^	0.32	71.0 ^a^	73.9 ^b^	75.6 ^c^	0.39					
	1		ALP				70.1 ^a^	71.5 ^ab^	74.1 ^c^	0.55					
	1		KAT				71.8 ^b^	76.4 ^d^	77.2 ^d^						
	2			73.3 ^a^	74.8 ^b^	0.32	72.5 ^a^	74.4 ^b^	75.3 ^b^	0.39					
	3			73.8 ^a^	74.8 ^b^	0.28	73.4 ^a^	74.5 ^b^	75.1 ^b^	0.34					
	4			75.9	76.1	0.35	74.6 ^a^	77.3 ^b^	76.5 ^b^	0.43					
	4		ALP				74.3 ^a^	78.2 ^d^	75.1 ^ab^	0.61					
	4		KAT				73.8 ^a^	76.4 ^bc^	77.9 ^cd^						
Propionate				14.1 ^b^	13.3 ^a^	0.25	15.1 ^c^	13.9 ^b^	12.2 ^a^	0.31	14.4 ^c^	14.0 ^bc^	13.0 ^a^	13.5 ^ab^	0.29
			ALP				15.3 ^d^	13.8 ^bc^	13.2 ^b^	0.44					
			KAT				14.9 ^cd^	13.9 ^bc^	11.1 ^a^						
	1			15.6 ^b^	13.2 ^a^	0.44	16.8 ^c^	14.3 ^b^	12.1 ^a^	0.54					
	2			14.0	14.0	0.46	15.0 ^b^	14.6 ^b^	12.3 ^a^	0.57					
	2		ALP				15.4 ^cd^	13.0 ^ab^	13.5 ^bc^	0.80					
	2		KAT				14.6 ^bcd^	16.2 ^d^	11.1 ^a^						
	3			13.3	12.8	0.36	14.6 ^b^	12.9 ^ab^	12.1 ^a^	0.44					
	4			13.5	13.4	0.34	14.5 ^b^	13.7 ^b^	12.3 ^a^	0.42					
Isobutyrate				0.40 ^b^	0.34 ^a^	0.015	0.50 ^c^	0.35 ^b^	0.27 ^a^	0.019	0.30 ^a^	0.35 ^a^	0.43 ^b^	0.41 ^b^	0.020
		ALF									0.36 ^c^	0.49 ^d^	0.51 ^d^	0.64 ^e^	0.035
		ALF+LES									0.33 ^bc^	0.29 ^abc^	0.47 ^d^	0.29 ^abc^	
		LES									0.22 ^a^	0.26 ^ab^	0.30 ^abc^	0.30 ^abc^	
			ALP								0.35 ^bc^	0.40 ^cde^	0.47 ^e^	0.38 ^cd^	0.029
			KAT								0.26 ^e^	0.29 ^ab^	0.38 ^cd^	0.44 ^de^	
Butyrate				10.5 ^b^	10.0 ^a^	0.21	10.2 ^a^	9.7 ^a^	11.0 ^b^	0.25	10.7 ^b^	10.5 ^b^	11.0 ^b^	9.0 ^a^	0.24
			ALP				10.4 ^bc^	10.1 ^abc^	11.0 ^c^	0.36					
			KAT				9.9 ^ab^	0.3 ^a^	11.0 ^a^						
	1			11.0	10.5	0.35	10.6	10.4	11.3	0.43					
	2			11.0 ^b^	9.9 ^a^	0.29	10.5 ^ab^	9.7 ^a^	11.2 ^b^	0.36					
	2		ALP				11.0 ^bc^	11.3 ^bc^	10.8 ^bc^	0.51					
	2		KAT				10.1 ^b^	8.2 ^a^	11.6 ^c^						
	3			11.0	10.9	0.35	10.4	10.9	11.6	0.42					
	4			9.1	8.8	0.37	9.1 ^b^	7.8 ^a^	10.0 ^b^	0.45					
Isovalerate				0.42 ^b^	0.31 ^a^	0.022	0.52 ^b^	0.32 ^a^	0.26 ^a^	0.027	0.29 ^a^	0.33 ^a^	0.42 ^b^	0.42 ^b^	0.030
			ALP				0.49 ^cd^	0.41 ^bc^	0.25 ^b^	0.039					
			KAT				0.55 ^d^	0.23 ^a^	0.17 ^a^						
	1			0.37 ^b^	0.22 ^a^	0.033	0.34 ^b^	0.32 ^b^	0.21 ^a^	0.041					
	2			0.42 ^b^	0.24 ^a^	0.027	0.47 ^b^	0.27 ^a^	0.25 ^a^	0.033					
	3			0.51	0.33	0.022	0.51 ^b^	0.46 ^b^	0.28 ^a^	0.027					
	4			0.37	0.47	0.069	0.74 ^b^	0.24 ^a^	0.29 ^a^	0.085					
	4		ALP				0.48 ^a^	0.29 ^a^	0.36 ^a^	0.122					
	4		KAT				1.00 ^b^	0.19 ^a^	0.22 ^a^						
Valerate				0.82 ^a^	0.78 ^b^	0.012	1.00 ^c^	0.77 ^b^	0.63 ^a^	0.015	0.79 ^a^	0.80 ^a^	0.86 ^b^	0.76 ^a^	0.016
			ALP				1.06 ^d^	0.77 ^b^	0.64 ^a^	0.021					
			KAT				0.94 ^c^	0.78 ^b^	0.63 ^a^						
	1			0.82	0.76	0.021	0.99 ^c^	0.81 ^b^	0.58 ^a^	0.026					
	2			0.82	0.78	0.024	0.96 ^c^	0.77 ^b^	0.67 ^a^	0.029					
	3			0.92	0.79	0.028	1.07 ^c^	0.84 ^b^	0.66 ^a^	0.028					
	3		ALP				1.22 ^d^	0.86 ^bc^	0.68 ^a^	0.039					
	3		KAT				0.93 ^a^	0.82 ^b^	0.63 ^a^						
	4			0.72	0.80	0.023	0.97 ^b^	0.68 ^a^	0.63 ^a^	0.028					
Acetate:propionate				5.38 ^a^	5.88 ^b^	0.109	4.91 ^a^	5.58 ^b^	6.40 ^c^	0.134	5.42 ^a^	5.52 ^ab^	5.82 ^b^	5.76 ^b^	0.124
			ALP				4.83 ^a^	5.54 ^ab^	5.77 ^b^	0.189					
			KAT				5.00 ^a^	5.62 ^b^	7.03 ^c^						
	1			4.73 ^a^	6.12 ^b^	0.192	4.31 a	5.39 ^b^	6.56 ^c^	0.231					
	1		ALP				4.17 ^a^	4.57 ^ab^	5.46 ^bc^	0.327					
	1		KAT				4.46 ^a^	6.22 ^c^	7.67 ^d^						
	2			5.44	5.60	0.198	4.89 ^a^	5.33 ^a^	6.34 ^b^	0.243					
	2		ALP				4.70 a	5.91 ^c^	5.71 ^bc^	0.343					
	2		KAT				5.09 ^abc^	4.74 ^ab^	6.98 ^d^						
	3			5.66	5.99	0.161	5.26 ^a^	5.85 ^b^	6.37 ^b^	0.201					
	4			5.69	5.83	0.151	5.19 ^a^	5.74 ^b^	6.34 ^c^	0.192					
Bacteria (×10^9^/mL)				8.64 ^b^	7.74 ^a^	0.214	8.79 ^b^	8.35 ^b^	7.44 ^a^	0.263	10.01 ^c^	9.11 ^b^	6.95 ^a^	6.70 ^a^	0.048
	1			10.79 ^b^	9.24 ^a^	0.358	10.33	9.82	9.89	0.438					
	2			9.66 ^b^	8.56 ^b^	0.378	9.72	8.31	0.463						
	2		ALP				8.93 ^b^	10.38 ^b^	9.68 ^b^	0.655					
	3		KAT				9.68 ^b^	9.06 ^b^	6.94 ^a^						
	3			6.79	7.12	0.359	7.58	7.21	6.06	0.440					
	4			7.34 ^b^	6.05 ^a^	0.302	7.96	6.64	5.48	0.370					
	4		ALP				9.60 ^c^	6.05 ^ab^	6.36 ^b^	0.523					
	4		KAT				6.32 ^b^	7.24 ^b^	4.60 ^a^						
Protozoa (× 10^5^/mL)				4.24 ^b^	3.74 ^a^	0.156	4.73 ^b^	3.77 ^a^	3.48 ^a^	0.191	3.19 ^a^	4.39 ^b^	4.15 ^b^	4.24 ^b^	0.173
Plasma															
UN (mg/L)				16.6 ^a^	18.8 ^b^	0.45	22.7 ^c^	18.3 ^b^	12.0 ^a^	0.55	15.4 ^a^	18.7 ^b^	20.9 ^c^	15.8 ^a^	0.59
			ALP				20.8 ^c^	16.6 ^b^	12.3 ^a^	0.77					
			KAT				24.6 ^d^	20.1 ^c^	11.7 ^a^						
TAC (μmol/L)				209	201	3.2	199	207	209	3.9	200 ^ab^	206 ^b^	196 ^a^	221 ^c^	3.8
		ALF									196 ^ab^	198 ^ab^	181 ^a^	221 ^cd^	6.7
		ALF+LES									199 ^ab^	219 ^cd^	201 ^b^	211 ^bc^	
		LES									205 ^bc^	199 ^ab^	203 ^b^	230 ^d^	

^1^ Breeds were Alpine doelings (ALP) and Katahdin ewe lambs (KAT); diets were 75% forage consisting of alfalfa hay (ALF), a 1:1 mixture of alfalfa and Sericea lespedeza hay (ALF+LES), and lespedeza hay (LES); periods 1–3 were 42 days in length and period 4 was 47 days; samples were collected at 4 h after feeding in week 5 of each period. ^2^ AMN = ammonia nitrogen; VFA = volatile fatty acids; UN = urea nitrogen; TAC = total antioxidant capacity. ^a,b,c,d,e,f^ Means within grouping without a common superscript letter differ (*p* < 0.05).

## Data Availability

Mean data are presented in tables.

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
