# Peer review of "Effects of Dietary Inclusion of Sericea Lespedeza Hay on Feed Intake, Digestion, Nutrient Utilization, Growth Performance, and Ruminal Fermentation and Methane Emission of Alpine Doelings and Katahdin Ewe Lambs"

_animals, 2022, doi:10.3390/ani12162064_

Round 1

Reviewer 1 Report

The paper is interesting and well structured, however is a duplicate of another paper already published by the same groups of researchers entitled “Effects of Dietary Inclusion of Sericea Lespedeza Hay on Behavior, Heat Energy, and Ruminal Methane Emission by Growing Alpine Doelings and Katahdin Ewe Lambs” https://www.ncbi.nlm.nih.gov/pmc/articles/PMC8104738/

The experiment was the same, only the behavior was investigated in the last paper and a more data concerning the growth performance and the nutrient utilization has been considered in the second one. However, for this reason I suggest to reject it.

Author Response

Reviewer #1

Comments:

  1. The paper is interesting and well structured, however is a duplicate of another paper already published by the same groups of researchers entitled “Effects of Dietary Inclusion of Sericea Lespedeza Hay on Behavior, Heat Energy, and Ruminal Methane Emission by Growing Alpine Doelings and Katahdin Ewe Lambs” https://www.ncbi.nlm.nih.gov/pmc/articles/PMC8104738/

The experiment was the same, only the behavior was investigated in the last paper and a more data concerning the growth performance and the nutrient utilization has been considered in the second one. However, for this reason I suggest to reject it.

Author response:

It appears that Reviewer #1 misunderstood that the data reported in this paper has been published as an article in the Journal of Animal Science.  The data presented in this manuscript have not been published as a full paper, but, rather there was an abstraction presentation at a conference.  Again, the abstract is published in the Journal of Animal Science and not as a full paper.  We hope that this is now clear.

Reviewer 2 Report

General:

I believe the manuscript is worthy of publication but a number of edits are needed to improve clarity and provide additional information.

The manuscript presents a lot of (almost excessive) data but there is little discussion of the results and explanations/comparisons.

The M&M need some additional information – how many animals/gates per pen?  If the same treatment was fed in all gates within the pen then the pen becomes the experimental unit – rather than the animal.  It seems, based on my interpretation of the text, (which is somewhat unclear) is what happened. 

Why 4 periods?  Were animals on the same treatment throughout the study or were the treatments changed as in a Latin square – this is not clear.  How many animals/pens per dietary treatment?

Although it is not clear (writing) I assume the sheep and goats were in different pens.

I don’t think it is every clearly noted how gross energy values were determined for feeds, urine, feces, etc.?

Provide compete chemical analysis of the two forages – alfalfa and les.

Inclusion of all the P values tables is probably unnecessary and just complicates the presentation of the actual data – which could be presented in a cleared form in the tables or in figures.  P values could be included in data table using footnotes.

Were individual feed ingredients sampled and analyzed or a sample of the complete mixed rations? 

Give specifics on the methods used to analyze the feeds, feces, etc for nutrients.

Tables are very difficult to read.

Specific:

Change “level” to “concentration” or “quantity” throughout to be more specific.

Page 2 – abstract              rephrase – “ Intakes of digestible and metabolizable energy and heat and retained energy were not …”

Table 4 and others – why not include the data for BW for ALP and KAT for each diet? Etc.

Author Response

General:

  1. I believe the manuscript is worthy of publication but a number of edits are needed to improve clarity and provide additional information. The manuscript presents a lot of (almost excessive) data but there is little discussion of the results and explanations/comparisons.

Author response:

Thank you for your valuable suggestions. We agree the manuscript is long, but it has a number of interaction effects that are useful to understand regarding the feeding of Lespedeza properly. We had described the results that were presented in tables, and the results with significant and important effects were explained in the Discussion section. Any specific comments on these aspects would be appreciated.

  1. The M&M need some additional information – how many animals/gates per pen?  If the same treatment was fed in all gates within the pen then the pen becomes the experimental unit – rather than the animal.  It seems, based on my interpretation of the text, (which is somewhat unclear) is what happened. 

Author response:

We mentioned in the manuscript “At most times, animals resided in six 6.1 × 5.6 m pens in an enclosed building that had a 6.1 × 1.35 m area with a concrete floor and a 6.1 × 4.25 m unpaved floor area.  The pens included Calan gate feeders (American Calan, Inc., Northwood, NH, USA) for individual feeding.”

“The pens and feeders were aligned in a row adjacent to one another.  With diets potentially differing in palatability, differences between breeds in size and behavior, and similar environmental conditions among pens, the same diet was fed to animals of the same breed in pens to avoid problems with attempts to gain access to feeders other than their own.” 

Calan gates allow one specific animal to consume feed from one specific and individual feeder.   In statistics, the experimental unit is the smallest unit to which a treatment is applied. Here treatments were three types of feed and treatments were applied to individual animals.  Feed intake by individual animals was determined.  All animals were kept in an enclosed barn having similar management conditions in all pens.  As noted above, if different diets were to have been fed in the same pen, then possibly because of differences in characteristics such as palatability it is likely that there would have been some ‘stealing’ of feed, resulting in inaccuracies in feed intake data.  We hope the reviewer is clear from our explanation.

There was no revision made for this comment.

  1. Why 4 periods?  Were animals on the same treatment throughout the study or were the treatments changed as in a Latin square – this is not clear.  How many animals/pens per dietary treatment?

Author response:

Variables were recoded in four periods to understand how the Lespedeza with increasing length of feeding can impact (for example, period x diet interaction, breed x diet x period interaction, etc) on the different measured variables. The experiment was conducted in a completely randomized design with a 2 x 3 factorial arrangement of treatment, not in a Latin square design. We have now more clearly described the experimental design in the revised manuscript.

  1. All animals were on the same treatment throughout the study.  How many animals/pens per dietary treatment?

Author response: Yes all animals were on the same treatment for 173 days. It was mentioned that “Twenty-four Alpine doelings (ALP; initial body weight and age of 25.3±0.55 kg and 10.4±0.11 mo, respectively) and 24 Katahdin ewe lambs (KAT; 28.3±1.02 kg and 9.6±0.04 mo, respectively) were used. The treatment arrangement was a 2 × 3 factorial, with the two species or breeds and three diets”. There were 8 animal per diet per breed and one pen per breed per diet. We have now clearly stated in the revised manuscript.

It was mentioned in the manuscript “The study began in January, 2019 and the duration was 173 days, with the first three periods 6 wk in length and the fourth 47 days.”  We have now added that “The animals consumed the treatment diets continuously for the total period of 173 days.” We hope it is now clear.

  1. Although it is not clear (writing) I assume the sheep and goats were in different pens.

Author response: In the manuscript, it was described that “With diets potentially differing in palatability, differences between breeds in size and behavior, and similar environmental conditions among pens, the same diet was fed to animals of the same breed in pens to avoid problems with attempts to gain access to feeders other than their own.” This means that goats and sheep in different pens and the three different diets were offered in different pens.

  1. I don’t think it is every clearly noted how gross energy values were determined for feeds, urine, feces, etc.?

Author response: It was mentioned that GE content of urine and feces was analyzed using a bomb calorimeter (Parr 6300; Parr Instrument Co., Inc., Moline, IL).  Feed was analyzed at a commercial laboratory that does not determine gross energy.  Thus, the following now has been stated:  “Fecal samples were analyzed for DM (100º C), ash [22], nitrogen (N; LecoTruMac CN, St. Joseph, MI, USA), neutral detergent fiber (NDF) with use of heat stable amylase [23] and containing residual ash, and gross energy (GE) using a bomb calorimeter (Parr 6300; Parr Instrument Co., Inc., Moline, IL).  Feed samples were analyzed at Custom Laboratory (Monett, MO, USA; customaglabs.com) for the same constituents by similar procedures, except for N that was determined by the Kjeldahl procedure [22].  Also, concentrations of acid detergent fiber (ADF) and acid detergent lignin (ADL) were determined [23].  Urine samples were analyzed for DM (lyophilization), N (LecoTruMac CN), and GE using the procedures stated above.” 

  1. Provide compete chemical analysis of the two forages – alfalfa and les.

Author response:  The total mixed diets were sampled rather than individual ingredients.

  1. Inclusion of all the P values tables is probably unnecessary and just complicates the presentation of the actual data – which could be presented in a cleared form in the tables or in figures.  P values could be included in data table using footnotes.

Author response: Due to large size of the tables, we were not able to present the p-values in the same tables. We believe the p-values would be useful to understand the results properly for this complex but interesting design.  It would not seem realistic to present P-values as footnotes.

  1. Were individual feed ingredients sampled and analyzed or a sample of the complete mixed rations? 

Author response:  Samples of the total mixed rations or diets were sampled, rather than individual ingredients.

  1. Give specifics on the methods used to analyze the feeds, feces, etc for nutrients.

Author response: It was described in the original manuscript. Some additional information has been included.

  1. Tables are very difficult to read.

Author response: This was a complex design with the effects of diet, breed, and period and their two- and three-way interactions. Thus, the tables are large as we needed to present interaction means when interactions were significant.

  1. Specific:

Change “level” to “concentration” or “quantity” throughout to be more specific.

Author response: We have revised the manuscript considering this comment.  However, it is conventional to use the word “level” as appear in the manuscript, with “quantity” or “amount” not always being appropriate.  Therefore, in a small number of places “level” was kept as such.

Page 2 – abstract              rephrase – “ Intakes of digestible and metabolizable energy and heat and retained energy were not …”

Author response: We have revised this sentence.

Table 4 and others – why not include the data for BW for ALP and KAT for each diet? Etc.

Author response: We presented main effect of breed and diet in all tables. For example, in Table 4.

Interaction

Breed

Diet

Period

Item2

Breed

Diet

ALP

KAT

SEM

ALF

ALF+LES

LES

SEM

1

2

3

4

SEM

BW (kg)

33.5a

44.4b

1.04

40.9

39.2

37.5

1.28

30.0a

36.6b

42.6c

47.5d

0.75

When interactions effects were significant, data were presented to show the interaction effects.
